# Fiber Bragg Sensors on Strain Analysis of Power Transmission Lines

**DOI:** 10.3390/ma13071559

**Published:** 2020-03-27

**Authors:** Janusz Juraszek

**Affiliations:** Faculty of Materials, Civil and Environmental Engineering, University of Bielsko-Biala, 43-309 Bielsko-Biala, Poland; jjuraszek@ath.bielsko.pl; Tel.: +48-338-279-191

**Keywords:** monitoring FBG, power transmission tower, civil engineering

## Abstract

The reliability and safety of power transmission depends first and foremost on the state of the power grid, and mainly on the state of the high-voltage power line towers. The steel structures of existing power line supports (towers) have been in use for many years. Their in-service time, the variability in structural, thermal and environmental loads, the state of foundations (displacement and degradation), the corrosion of supporting structures and lack of technical documentation are essential factors that have an impact on the operating safety of the towers. The tower state assessment used to date, consisting of finding the deviation in the supporting structure apex, is insufficient because it omits the other necessary condition, the stress criterion, which is not to exceed allowable stress values. Moreover, in difficult terrain conditions the measurement of the tower deviation is very troublesome, and for this reason it is often not performed. This paper presents a stress-and-strain analysis of the legs of 110 kV power line truss towers with a height of 32 m. They have been in use for over 70 years and are located in especially difficult geotechnical conditions—one of them is in a gravel mine on an island surrounded by water and the other stands on a steep, wet slope. Purpose-designed fiber Bragg grating (FBG) sensors were proposed for strain measurements. Real values of stresses arising in the tower legs were observed and determined over a period of one year. Validation was also carried out based on geodetic measurements of the tower apex deviation, and a residual magnetic field (RMF) analysis was performed to assess the occurrence of cracks and stress concentration zones.

## 1. Introduction

Power transmission lines extend over hundreds of kilometers. The collapse of one or more towers may have serious consequences, resulting in a local shutdown or even a blackout. Therefore, the systematic testing, maintenance and repair of tower elements are necessary. Unfortunately, power line operators often fail to do so. This especially applies to older structures which have been in use for over 30 years. The repair or replacement of a tower structure element is difficult because it is usually done manually or using a small winch. The tower’s location on a steep mountainside or an island surrounded by water often creates an additional difficulty.

Periodic testing is performed by measuring deformation or the tower apex deviation. An interesting analysis of a high-voltage power line tower structure deformation is presented in [1]. The analysis concerned towers (pylons) with a height of 28–32.2 m supporting a double 110 kV power line. The structure was made of S235 steel. The measurements were performed using a new FOTON photogrammetric system [2,3,4]. The biggest measured deviation in the pylon apex totaled 480 mm. The downside of the solution is that the photographing device should be perpendicular to the tested object and far away from it. Moreover, fog and poor lighting conditions can render the measurement impossible. Such measurements can be made periodically but not continuously, and they are troublesome in practice. Usually, they are performed once every few years and continuous measurements are impossible. A new 3D photogrammetric technique of measuring the tower deviation was introduced by Xiao et al. [5]. Special markers were used and photographs were taken as towers were subjected to loads. However, this method requires a high computing power to determine the deformation of tested elements based on a series of photographs. Another interesting method for automatic supervision is the efficient extraction and classification of three-dimensional (3D) targets of electric power transmission facilities based on regularized grid characteristics computed from point cloud data acquired by unmanned aerial vehicles (UAVs) [6]. A special hashing matrix was constructed first to store the point cloud after noise removal by a statistical method which calculated the local distribution characteristics of the points within each sparse grid. Next, power lines were extracted by the neighboring grids’ height similarity estimation and linear feature clustering. After that, by analyzing features of the grid in the horizontal and vertical directions, the transmission towers in candidate tower areas were identified. Unfortunately, this method does not enable a precise assessment of the tower deviation or strains in the tower legs. The system can be useful for monitoring long rows of power line towers and making a preliminary assessment of the tower apex deviation. An analysis of tower failures was also conducted by Davies [7]. It was concluded that 31% of tower failures were due to design errors, 29% were caused by ice, strong winds accounted for 19% and other causes were responsible for 11%. The towers analysed herein have been in use for more than 70 years, which confirms that they are not burdened with design errors—these would have revealed themselves by now. The American (ASCE) and British power transmission tower design standards were analyzed by Rao [8]. The author assessed them as very conservative with regard to experiments. The forces determined according to nonlinear theories turned out to be smaller than originally assumed, which may lead to the oversizing of structures.

Numerical as well as numerical-and-experimental simulations of power line towers were also performed, which were described, for example, in the works of Yin [9] and Lam and Yin [10] on experimentally verified numerical simulations of towers using the finite element method (FEM). One of the methods of monitoring the state of power line towers is vibration analysis [11]. This idea concerns measurements of the response frequency function of the tested tower. If damage occurs, the resonance peak is shifted. If the tower loses stiffness due to cracks, the frequency of vibrations decreases. Such changes can be considered indications of damage. However, the precise localization and identification of damage require not only accurate numerical models to simulate the tower vibrations, but also adequately high computing power. The significant researchers in this area are Al-Bermami and Kitiponrchai [12]. They simulated the occurrence of damage in two towers in Australia. A review of the design practices of the 1990s in this regard is presented in [13]. Considering current climate changes, the infrastructure of power transmission towers should be strengthened using appropriate stiffening devices [14]. At a later stage of the research, Al-Bermami applied a nonlinear methodology of anticipating the occurrence of structural damage in towers [15]. The above-mentioned impact of wind is one of the most typical loads. It may cause power lines to vibrate and damage to towers. Because weather conditions are now increasingly dramatic, the problem is becoming more and more serious. Interesting findings are presented by Frei et al. [16], who investigated the structural stability of power transmission lines. They based the stability assessment on the static analysis and the dynamic analysis. It follows that low modal frequencies become smaller before instability is reached (before a failure occurs), which may be an early warning sign.

An analysis of damage to a power transmission tower due to strong wind was conducted in [17,18]. Both static and dynamic simulations were performed. The analysis covered multi-member towers with a total height of 59 + 17.5 = 76.5 m. The critical strength values were compared to the American (ASCE) and the British standard and to the DL/T 5154–2012 guidelines [19,20].

Depending on the standard, they are included in the range of 250–270 MPa for the tower legs and 100–140 MPa for the diagonal elements. The results of the analyses indicate that more emphasis should be put on the design of diagonal elements. The analyzed tower was designed using steel angle sections. A combination of an interesting analysis of a full-scale test and a numerical simulation of a power transmission tower with a height of 46.05 m and a 7.14 m × 7.14 m base under the impact of wind is presented in [21,22,23].

The full-scale test is the most reliable way to reflect the power transmission tower’s mechanical properties. Obviously, the experimental cost is very high and the test is only possible for new tower types [24,25,26].

Ma et al. [27] proposed using an FBG sensor as a small aerometer to measure the wind speed. The essence of the sensor operation was the bending of the sensor’s active element under a wind load. The sensor deformation was proportional to the wind speed. The lowest measurable wind speed totaled 3 m/s and the measurement error was smaller than 0.1 m/s. The device worked very well in difficult environmental conditions. FBG sensors are mainly used as strain–tension sensors. Most of the FBG sensors available on the market enable the connection of a few dozen sensors, each with a different wavelength (differing by 5 nm), to one telecommunication optical fiber. The use of optical multiplexes makes it possible to connect and build measurment networks with the capacity of several thousand FBG sensors. The important thing is that optical fibers are resistant to electromagnetic noise, which is crucial in measurements involving high voltage. The advantage of the FBG technology is the possibility of using high sampling frequencies of up to 2 kHz, which enables the analysis of dynamic as well as long-lasting quasi-static processes. The works present interesting investigations of deformations of complex metal structures using FBG sensors, as well as an innovative combination of deformation measurement techniques based on optical fibers and the Residual Magnetic Field method, which make it possible to identify stress concentration zones in a steel structure in advance [28,29]. The issue presented in this paper concerns the diagnostics of power transmission towers which have been in service for about 70 years and are located in especially difficult geotechnical conditions. The towers are truss structures with a height of up to 32 m. In the literature, there are no works on the diagnostics of this type of tower located in especially difficult terrain conditions, where deformation processes occur over a long period of time. The author proposes the use of appropriately designed FBG sensors enabling the analysis of strains in the tower legs with a parallel use of the residual magnetic field (RMF) method which makes it possible to detect the structure stress concentration zones.

## 2. Materials and Methods

A system monitoring the deformation of the structure of power transmission towers was built. The system was based on fiber Bragg grating (FBG Sylex, Bratislava, Slovakia) strain sensors. The essence of the solution was to introduce FBG sensors in certain points of the tower structure as a kind of “nervous system”. In the case of supporting structures of power transmission lines, the appropriate places were those with the highest bending moment and the biggest compressive stresses. The places were pre-determined using numerical simulations and they were located at the tower base. The entire optical fiber system intended for the strain-and-stress state measurement was duly calibrated. Special grips had been designed earlier to make it possible to fix the strain sensor to the selected tower leg repeatedly. The FBG strain sensor is presented in Figure 1.

Based on preliminary testing, the measuring base L = 1000 mm was adopted. Using special grips, the sensor axis was placed in the center of gravity of the tower leg angle section.

The measuring part of each strain sensor was a Bragg grating with a specific wavelength embedded in an optical fiber. One sensor was used for each tower leg, i.e., there were four sensors per tower. Due to the optical interrogator traceability, a difference of at least 5 nm between the wavelengths of each Bragg grating had to be kept. A change in strains in the tested structural element, i.e., in the tower leg, caused a change in the wavelength in the Bragg grating. This relation is described by the following formula (1):(1)$$\Delta \varepsilon =\left[\frac{\lambda_{a}-\lambda_{0}}{\lambda_{0}}-B\;\left(T_{a}-T_{0}\right)\right]\;A^{\left(-1\right)}$$
where:
 Δε strain shift;$\;\lambda_{a}$ actual strain wavelength;$\;\lambda_{0}$ initial strain wavelength;$\;T_{a}$ actual temperature;$\;T_{0}$ initial temperature;*A* and *B* calibration parameters. *A* = 7.8648540 × 10^−7^ [με^−1^]; *B* = 5.99813502 × 10^−6^ [°C].

Another important factor was to select an appropriate initial tension of the sensor. The sensor’s initial tension was set precisely using right-and-left nuts with a fine thread. Apart from the dedicated optical fiber sensors, the system included a 2 kHz FBG-800 optical interrogator (Fibre Optic Sensing & Sensing Systems–FOS&S, Geel, Belgium), a recorder, special software, a multiplier and telecommunication fibers. An option was also possible with a wireless transfer of measurement results from the interrogator. The constructed optical fiber system intended for the measurement of strains arising in the tower legs was first tested in laboratory conditions. The tests consisted of setting the known force and strain values to a fragment of a tower leg cut-out together with the FBG sensor fixed on it. The fixing manner ensured that the cut-out response was identical with that of the actual tower leg. Considering the required spacing between the traverses of the strength-testing machine with Accuracy Class 0.5, the tests were performed in the Research and Supervisory Center of Underground Mining. The diagram of the tension and compression test rig is presented in Figure 2. Special grips were designed to fix the angle section in the jaws of the strength-testing machine and to enable the angle section buckling in relation to the axis characterized by the lowest value of the second moment of inertia. In the first stage, tests calibrating the measuring system were carried out in the following sequence:(a)testing the tower structural elements together with installed FBG SC-01 sensors with a measuring base of 1000 mm. The testing was carried out on the tower leg fragment in the form of 80 mm × 80 mm × 8 mm and 75 mm × 75 mm × 6 mm angle sections with special grips fixing the optical fiber sensors and enabling the appropriate initial tension;(b)for the assumed constant 5 kN increment in force set by the strength-testing machine, strains corresponding thereto and established experimentally using the FBG SC-01 sensor were determined.

The testing results proved unequivocally that the optical fiber sensor and the method of fixing the sensor enabled a correct measurement of strains arising in the tested angle section of the tower leg.

The angle section with the attached grips and the installed FBG sensor was subjected to a tensile force and a compressive force included in the same range as in the tower leg real conditions. The results of the testing are illustrated by the load-strain characteristic plotted for the tested leg (cf. Figure 3).

For every 5 kN increment in force, there was an increment in strain totaling 23 µstrain. It can be observed that constant increments in load result in constant increments in strain. The testing results prove unequivocally that the optical fiber sensor and the method of fixing the sensor to the tower leg angle section enabled a correct measurement of strains in the case of both tensile and compressive forces.

Using the results of the laboratory testing, performed on the strength-testing machine (XB-OTS-600, 100 kN, Dongguan Xinbao Instrument Co., Ltd., Guangdong, China), of the angle sections used as the power line tower supporting elements and of the designed grips with the FBG sensors fixed to them, the angle sections were verified and re-designed to obtain the final form, as presented in Figure 4.

The change in design made the assembly of the grips easier and ensured an appropriate and constant initial tension of the sensors at a level of about 600 µstrain.

## 3. Results of Strain Testing of High-Voltage Power Line Towers

The strain testing results were composed of two parts. The first presented the results of measurements of strains arising in the tower legs under a simulated short-term lateral load of 500 N. The load was applied in the direction of the tower two axes (I, II) and along its diagonals (III). A diagram illustrating the loading is presented in Figure 5.

The load applied to the tower in direction II resulted in compressive strains of −14 µstrain in legs 1 and 2. A tensile strain of 10 µstrain arose in the legs on the tower’s opposite side. The strain measurement results are presented in Figure 6a,b.

This meant the occurrence of tensile stresses of 2.1 MPa and compressive stresses of about −2.82 MPa. If a load was applied in direction III, the values of strain for leg XP1 total −19 µstrain, and for the leg located diagonally, 14 µstrain. The highest strain values were recorded if the tower was loaded along the axis in relation to which the tower apex was shifted. The recorded strain values totaled 21 µstrain (cf. Figure 7), which, based on physical relations, meant a tensile stress of about 4.5 MPa.

The second part of the testing concerned the observation of strains arising in selected legs of the tower over a period of one year. In the case of the legs of tower A, located on an island in a gravel mine area, strains rose in the winter period to 230–270 µstrain and then returned, in the case of leg 1, to the initial level of 150 µstrain. In the case of tower B, the course of the changes was similar, but the strain values did not return to the initial state and kept at the level of 230 µstrain. The tower leg was shortened, which meant that the stresses arising in the two legs were compressive. The strain values for the other two legs were similar but with the opposite sign, which meant that there was tension. By determining real strain values in the tower legs, it was possible to define the performance of the tower foundation. The tower rotation axis (direction II) at the tower base is marked in Figure 5. It passed between the legs subjected to compression and those subjected to tension. The hypothesis was also confirmed by the measurements of the tower apex deviation (cf. subsection on validation). Using physical relations (Hooke’s law), the values of stresses arising in the tower’s individual legs were determined. They are presented in Figure 8.

The changes in stresses were correlated with the changes in temperature (cf. Figure 8). Analyzing the temperature history during the observation, it can be noticed that stresses in the tower varied with changes in temperature. A drop in temperature involved a rise in the compressive stress value in the tower legs (to observation point 11). The rise in temperature that occurred in the post-winter period caused an increase in compressive stresses. In leg 1, stresses were increased by about 20 MPa, whereas in leg 2 the rise in compression was higher and totaled about 25 MPa. In summer, temperatures were higher and according to the observation results, stresses in leg 1 remained at the same level, whereas compressive stresses in leg 2 dropped to the value of about −50 MPa. The highest calculated value of compressive stress totaled 60 MPa. The stresses were compared to allowable stress values for this type of tower structure, which were defined precisely in [17]; they range from 150 to 250 MPa.

The tower located in the mountains did not display such regularity (cf. Figure 9).

This is especially visible in the case of leg 2, for which the relations were the opposite, i.e., a rise in temperature in the initial period involved a drop in stresses by 15 MPa and a change in the nature of the stress, whereas a temperature drop changed the sign of stresses and increased them to the value of 5 MPa. It can be seen that as leg 4 was subjected to tension, leg 4 was compressed. It can be supposed that this is due to the landform features and# the fact that the terrain was unstable and wet. Most probably, the tower foundation rotated in relation to the axis along one of the tower base diagonals. Moreover, each of the tower legs had a separate foundation. The tower’s hypothetical rotation at the tower base is presented in Figure 5 (direction III). In such a situation, one side of the tower structure was subjected to tension and the other to compression. This was also confirmed by geodetic measurements of the tower apex.

## 4. Validation

Validation by means of geodetic measurements was carried out on the tower apex deviation from the plumb line. The method of direct projection onto a leveling staff placed on the tower foundation was applied. A Leica TCR 407 tacheometer (LEICA, Wetzlar, Germany) and a measuring staff were used. The measurements revealed a tower axis deviation from the plumb line of 3 and 13.5 cm, respectively, in the direction parallel (component X) and perpendicular (component Y) to the course of the 110 kV high-voltage power transmission line (Figure 10).

The results of the measurements of the tower deviation confirmed the results of the measurements of strains in the tower legs. They indicated that the tower structure inclined towards legs 2 and 3 and caused higher compressive stresses.

The tower legs were also tested using the residual magnetic field (RMF) method. The method is described in [28,29] and makes it possible to detect stress concentration zones in ferromagnetic elements, such as steel, used to build the tower legs. Stress concentration zones occur in areas where the magnetic indicator—the gradient of the normal and the tangential component along the measuring path—reaches a value exceeding 10 [(A/m)/mm.] No such areas were found in the case of the tower legs under consideration. Moreover, the method enables the detection of fractures and microcracks. In these particular areas, the residual magnetic field’s normal component becomes zero. Despite the long in-service time, the RMF testing did not detect any cracks in the tower leg. The distribution of the gradient of the tangential component magnetic field along the tower A leg is presented in Figure 11.

## 5. Conclusions

The conducted research proved an essential influence of difficult geotechnical and environmental conditions of the power line tower foundation on the strain state in the tower legs. The testing of supporting structures of high-voltage power transmission lines by means of optical FBG systems provided a lot of essential information about the real values of strains and stresses occurring in the legs of the analyzed towers of overhead power transmission lines. This is of particular importance in the case of lines which have been in use for many years. The obtained information may also contribute to an improvement in the operating safety of power line towers. The presented fiber-based system intended for the measurement of strains enabled dynamic measurements (horizontal load) as well as long-term measurements. The results of the strain-state measurements of the tower legs were confirmed by geodetic measurements of the tower apex deviation from the plumb line. Using various research methods, e.g., combining the FBG system to measure strains with the assessment of the tower’s strain-and-stress state using the RMF method, made it possible to evaluate the performance of a tower structure in very difficult geotechnical conditions.

## Figures and Tables

**Figure 1 materials-13-01559-f001:**
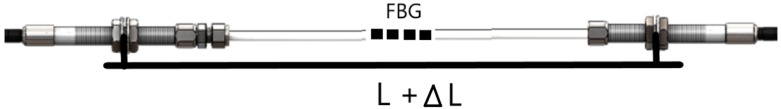
The fiber Bragg grating (FBG) strain sensor.

**Figure 2 materials-13-01559-f002:**
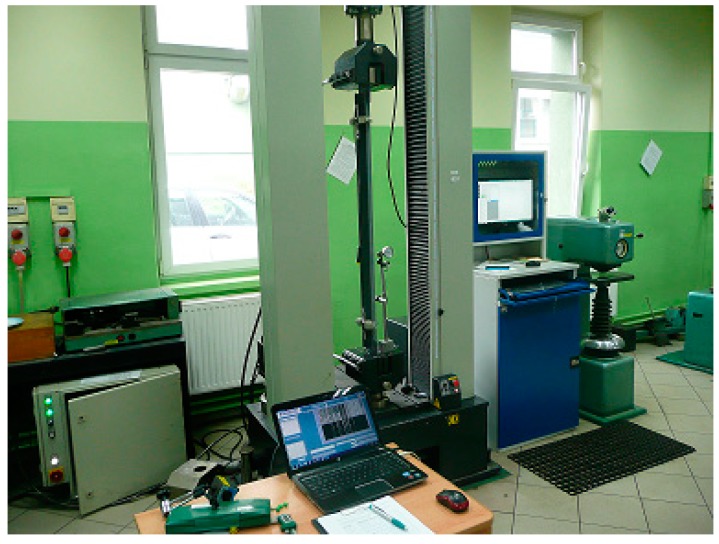
Laboratory stand intended for the calibration of the strain-state measuring system.

**Figure 3 materials-13-01559-f003:**
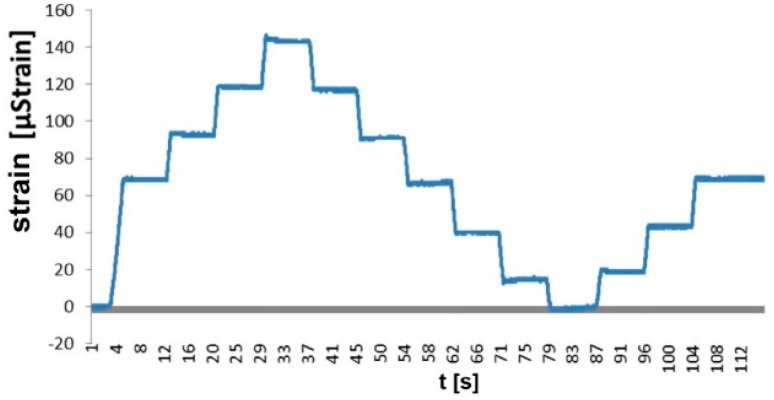
Calibration of the optical fiber transducer on the strength-testing machine.

**Figure 4 materials-13-01559-f004:**
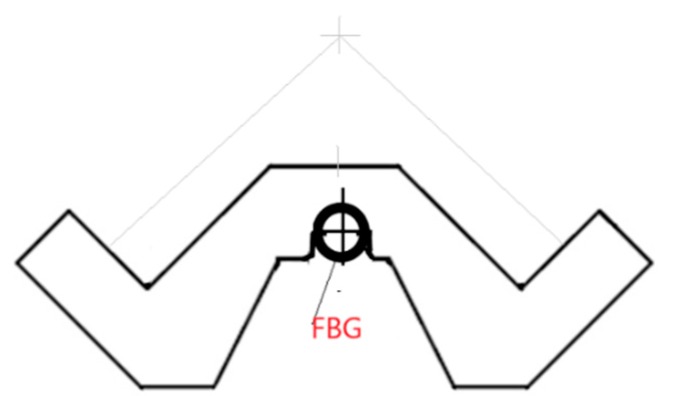
Designed grips for the fixing of FBG SC-01 and SC-02 sensors.

**Figure 5 materials-13-01559-f005:**
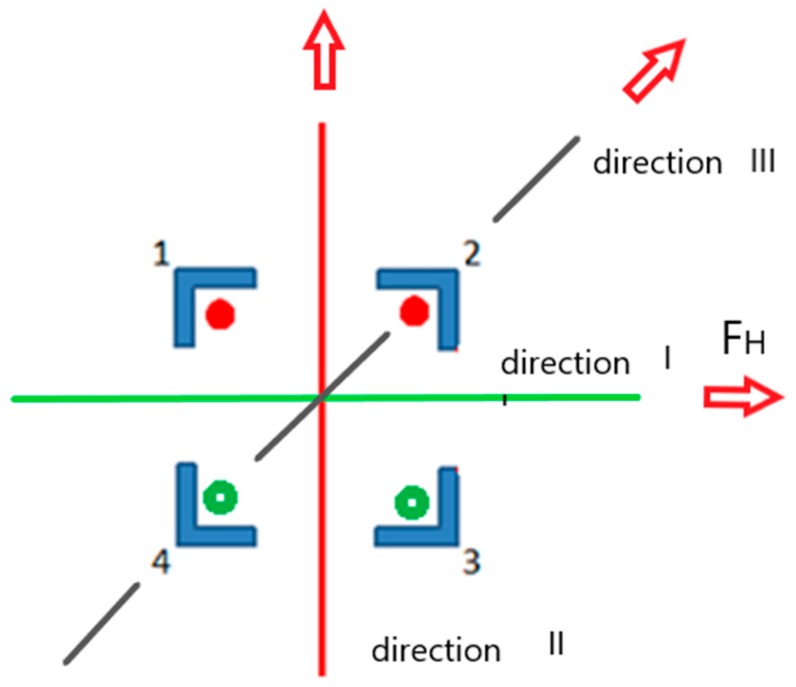
Direction of the load force in tower A.

**Figure 6 materials-13-01559-f006:**
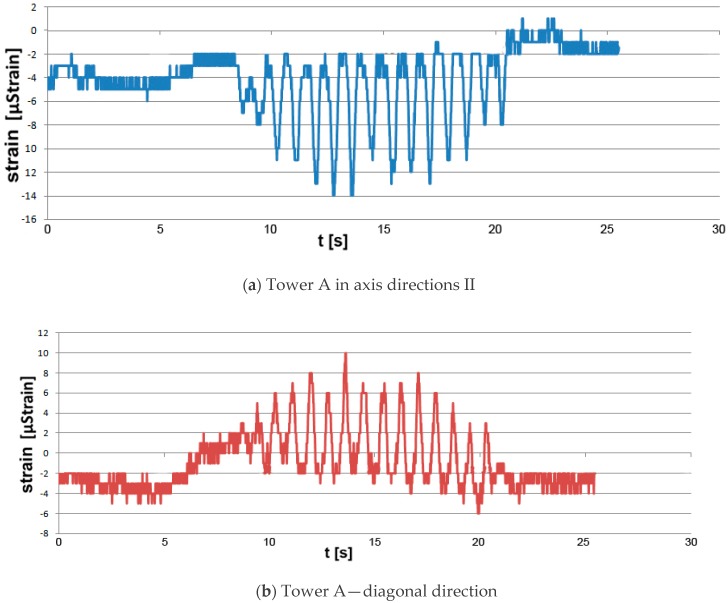
Strains in the legs of tower A under a lateral load: (**a**) in the tower axis direction, (**b**) in the tower diagonal direction.

**Figure 7 materials-13-01559-f007:**
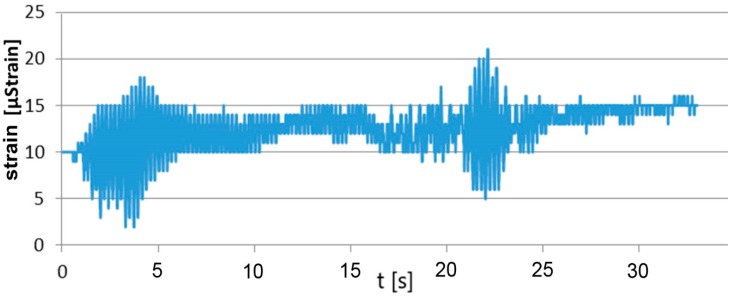
Highest strain values under a lateral load.

**Figure 8 materials-13-01559-f008:**
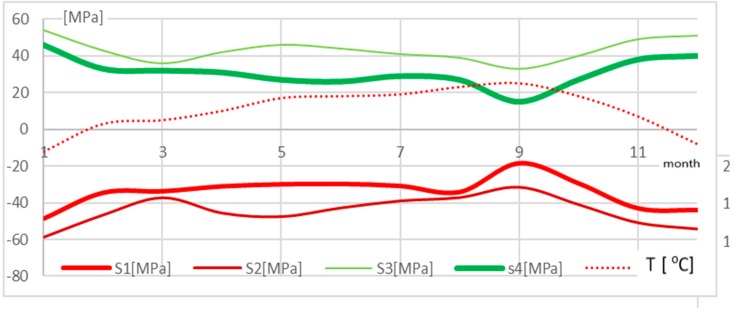
Stress and temperature distribution in the legs of tower A located on an island in a lake.

**Figure 9 materials-13-01559-f009:**
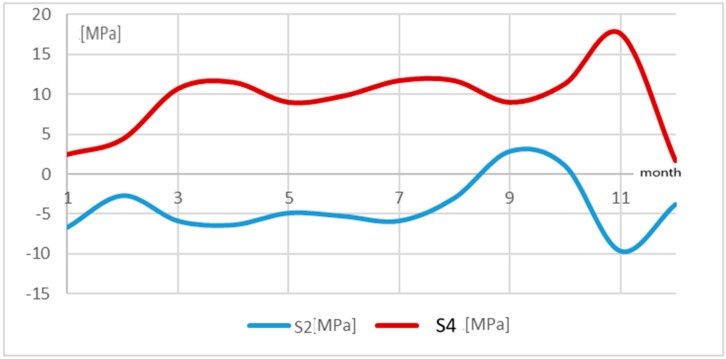
Stress and temperature distribution in the legs of tower B located in the mountains.

**Figure 10 materials-13-01559-f010:**
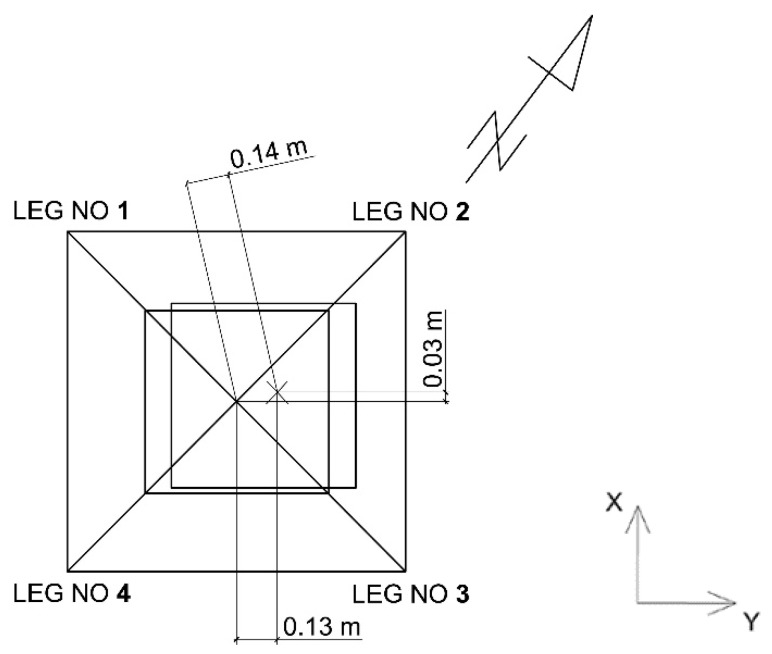
Tower A rotation and translation.

**Figure 11 materials-13-01559-f011:**
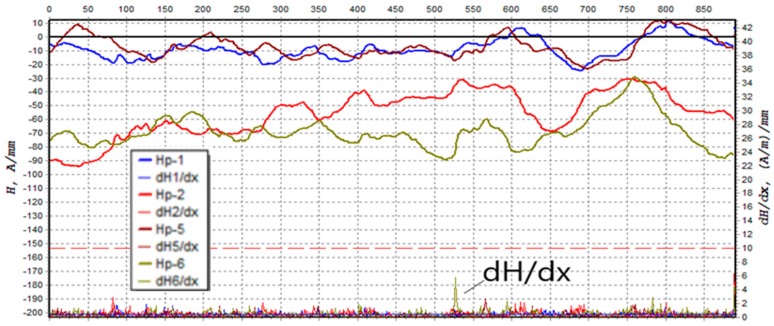
Distribution of the gradient dH/dx tangent component of magnetic field.

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
