# Peer review of "Fiber Bragg Sensors on Strain Analysis of Power Transmission Lines"

_materials, 2020, doi:10.3390/ma13071559_

Round 1
Reviewer 1 Report
The author describes an FBG-based sensing setup for analyzing strain levels of power transmission lines. The research has merit, but a few points must be addressed before final consideration.
- Descriptions in Fig. 4, Fig. 7, and Fig. 12 are too small.
- In Fig. 7a, there is a shift between the strain level in t=0 and t=25s. Please comment on it.
- Please add other time values to the horizontal axis of the plot in Fig. 8. In the current version, there are only values t=0 and t=30s.
- Axis in Fig. 9 doesn’t have any description.
- Please clarify how the temperature cross-sensitivity in the FBGs are considered in the measurements.
- Please further develop the analyses in Section 4 (Validation). Is it possible to draw a quantitative comparison between the methods?
Author Response
Replies to comments of Reviewer 1
Re: 1. The descriptions in the figures have been enlarged.
Re: 2. The jump in strain between t=0 and t=25s results from the structure vibrations; it decreases in a longer time of observation.
Re: 3. More time values have been added to the horizontal axis.
Re: 4. The axis in Fig. 9 is now properly described.
Re: 5. The temperature cross-sensitivity is taken into account by precise strain calibration in different temperature conditions in the metrological laboratory and introduction of an additional passive optical fibre loaded only thermally.
Re: 6. The author is now working hard to draw a quantitative comparison between the methods.
Reviewer 2 Report
The paper reports the strain analysis of power transmission lines using FBG sensors. The application of FBG strain analysis in towers located in remote areas is interesting. The author can consider the following comments/suggestions to bring the article to an acceptable level.
Section 2: Materials & Methods
- Where is Figure 1? The text refers to Figure 2 without citing Figure 1.
- The caption for Table 1 is missing.
- Where are the footnotes cited in Table 1? (**1 & **2)
Section 3: Results
- The notation for directions mentioned in the text and Figure 6 does not match. Line 189: the principal directions have been labelled as I and III while on the figure they are I and II.
- Line 193-4: The measured tensile and compression strain along legs are 10 and 14 micro-strain, respectively. However, the tensile stress (2.82 MPa) is higher than the compression stress (2.1MPa). Please add details of your stress calculation and explain why tensile stress is higher, which does not seem to be correct.
- Line 196-7: Why the measured strains in direction 3 at both leg XP1 and the one located diagonally are in compression? However, in Line 200, tensile stress has been reported.
- To which tower (A or B) does Figure 7 refer? Please reword the caption.
- What are the axis titles for figure 9? The unit for temperature is not C rather oC. Please correct.
- Line 228: Where is the "post-winter period" in Figure 9?
- Line 232: "The highest recorded value of compressive stress totalled 60 MPa". Based on the paper, the values of stress have not recorded rather calculated. Please reword.
- What is the effect of residual stresses on the measured strain for tower B? (Line 238)
- Line 239: Where do you refer to in this sentence: "It can be seen that as leg 4 was subjected to tension, leg 3 was compressed." The stress values for leg 3 and 4 have not shown in Figure 10.
Section: Validation
- It is not clear the validation of the measured strain has been conducted with respect to tower A or B (Line 249-50). Same as line 265.
Section: Conclusions
- It has been claimed that "The conducted research proves an essential influence of difficult geotechnical and environmental conditions of the power line tower foundation on the strain state in the tower legs." Since there is no geotechnical info in the article, what is your justification for the claimed statement?
There are a few typos on the paper which have been highlighted in the attached file. Proofreading is recommended.

Author Response
- Where is Figure 1? The text refers to Figure 2 without citing Figure 1.
Re:1. There is no Fig. 1. The numbering of the figures has been corrected.
- The caption for Table 1 is missing.
Re: 2. An appropriate caption has been introduced for Table 1.
- Where are the footnotes cited in Table
1? (**1 & **2)
Re: 3. The footnotes in Table 1 have been removed and the table has been corrected.
Section 3: Results
- The notation for directions mentioned in the text and Figure 6 does not match. Line 189: the principal directions have been labelled as I and III while on the figure they are I and II.
Re: 1. The direction marking in Fig. 6 has been corrected.
- Line 193-4: The measured tensile and compression strain along legs are 10 and 14 micro-strain, respectively. However, the tensile stress (2.82 MPa) is higher than the compression stress (2.1MPa). Please add details of your stress calculation and explain why tensile stress is higher, which does not seem to be correct.
Re: 2. There was a mistake in lines 193-4: the stress and strain values were transposed. The tensile stress value should have been 2.1 (instead of 2.82) and the correct value for compression should have been 2.82 (instead of 2.1), which has now been corrected.
The values were calculated based on physical relations according to Hooke’s law: σ = E * ε, where ε is strain and E is Young’s modulus = 2.1e5MPa or 210000MPa.
- Line 196-7: Why the measured strains in direction 3 at both leg XP1 and the one located diagonally are in compression? However, in Line 200, tensile stress has been reported
Re: 3. An editing error. The strains of leg 4 are tension-related and total 14. The strains described in line 200 are related to the highest value recorded during all the tests.
- To which tower (A or B) does Figure 7 refer? Please reword the caption.
Re: 4. Fig. 7 refers to tower A. The caption has been modified.
- What are the axis titles for figure 9? The unit for temperature is not C rather oC. Please correct.
Re: 5. The axis titles and the correct temperature unit have been added to Fig. 9.
- Line 228: Where is the "post-winter period" in Figure 9?
Re: 6. The post-winter period in Fig. 9 is the area between month 4 and month 6 (on the horizontal axis).
- Line 232: "The highest recorded value of compressive stress totalled 60 MPa". Based on the paper, the values of stress have not recorded rather calculated. Please reword.
Re: 7. Following the Reviewer’s suggestion, “recorded” has been replaced with “calculated” (values).
- What is the effect of residual stresses on the measured strain for tower B? (Line 238)
Re:8. This will be found in further studies, which are now being conducted.
- Line 239: Where do you refer to in this sentence: "It can be seen that as leg 4 was subjected to tension, leg 3 was compressed." The stress values for leg 3 and 4 have not shown in Figure 10.
Re: 9. An editing error: It should have been “leg 2” (not 3). Corrected.
Section: Validation
- It is not clear the validation of the measured strain has been conducted with respect to tower A or B (Line 249-50). Same as line 265.
Re: 1. The following sentence has been added: The validation was carried out for tower A.
Section: Conclusions
- It has been claimed that "The conducted research proves an essential influence of difficult geotechnical and environmental conditions of the power line tower foundation on the strain state in the tower legs." Since there is no geotechnical info in the article, what is your justification for the claimed statement?
Explanation of geotechnical conditions.
The tower A is located on the shore of a man-made lake. The whole steel structure is founded on a massive concrete block. The subsoil is granular and consists of gravel mainly. The soil bearing capacity varies over time due to gravel mining close to the foundation and possible fluctuations in the water level. The subsoil condition should be considered as exceptionally difficult.
Dredgers working nearby may cause gravel loosening, which leads to a reduction in the soil bearing capacity and lateral resistance close to the foundation block. The subsoil weakening is probably the main reason for the observed displacement of the steel structure.
The tower B is located on a steep natural slope built of Carpathian flysch. The flysch formation is characterized by a layered structure of stiff rock, such as sandstone and weaker claystone, mudstone and shales. In addition, water outflows are found on the slope. The geological structure and the hydrological condition indicate the possibility of the soil-mass movement. The subsoil condition should be considered as exceptionally difficult.
The steel structure is founded on different shallow concrete footings. The foundation type, the soil inhomogeneity and the hydrogeological condition are probably the main reasons for the observed displacement of the steel structure.
Round 2
Reviewer 2 Report
The author has implemented / replied to all comments.